# Technical note: Finite element formulations to map discrete fracture elements in three-dimensional groundwater models

Rob de Rooij

Water Institute, University of Florida, 570 Weil Hall, PO Box 116601, Gainesville, FL-32611-6601, USA

*Correspondence to*: Rob de Rooij (r.derooij@ufl.edu)

**Abstract.** Typically, in finite element groundwater models, fractures are represented by two-dimensional triangular or quadrilateral elements. When embedded in a three-dimensional space, the Jacobian matrix governing the transformation from the global three-dimensional space to the local two-dimensional space is rectangular and thus not invertible. There exist different approaches to obtain a unique mapping from local to global space even though the Jacobian matrix is not invertible.
These approaches are discussed in this study. It is illustrated that all approaches yield the same result and may be applied to curved elements. The mapping of anisotropic hydraulic conductivity tensors for possibly curved fracture elements is also discussed.

## 1 Introduction

The finite element method is well-suited for accommodating fractures in groundwater models. Typically, fractures are represented by discrete two-dimensional elements and these fracture elements can be embedded within a three-dimensional continuum consisting of three-dimensional elements. For example, within a tetrahedral mesh, fractures can be embedded by using triangular elements such that each triangle corresponds to a face shared by two adjacent tetrahedral elements. Similarly, quadrilaterals can be embedded within a hexahedral mesh. Indeed, such discrete-continuum models with embedded fractures
are routinely applied (Blessent et al., 2009; Blessent et al., 2011; Li et al., 2020; Watanabe, 2011).

A key component in the finite element method is the mapping of the gradient matrix from local to global space, where the global space is typically defined by a standard orthogonal coordinate system. The local space within a finite element can be curvilinear and has the same dimension as the element itself. If the global space has the same dimension as the local space, then the mapping is defined by the inverse of the Jacobian matrix. However, in the case of two-dimensional fractures embedded
in a global three-dimensional space, the Jacobian matrix is non-square and thus not invertible (Juanes et al., 2002; Perrochet, 1995). A couple of different techniques enable a mapping from two-dimensional local to three-dimensional global space.

A first approach is based on using contravariant base vectors and the contravariant metric tensor (Kiraly, 1985; Cornaton et al., 2004; Juanes et al., 2002; Perrochet, 1995). This approach requires some understanding of tensor calculus and the few

studies that describe this approach refer to mathematical textbooks for more details. Nonetheless, this approach yields a rather simple expression for the mapping and is directly applicable to curved elements.

A second approach uses the right Penrose-Moore inverse of the Jacobian matrix. As shown in this study, the derivation of this pseudo-inverse is relatively straightforward. Within the field of finite elements, the left Penrose-Moore inverse has been applied for the reverse mapping from a three-dimensional global space to a two-dimensional local space (Rognes et al., 2013). One study mentions the pseudo-inverse for mapping finite elements to higher dimensions (Reichenberger, 2004), but only within the context of non-curved elements and without much further detail.

A third approach is to introduce an intermediate mapping to an orthonormal two-dimensional space tangent to the fracture space. The Jacobian of such a mapping is invertible. A matrix of directional cosines is used for a subsequent mapping to the global space. This approach is widely used, and the available literature is quite detailed (Diersch et al., 2005; Watanabe, 2011; Kolditz and Glenn, 2002). However, the approach as discussed in available literature is only applicable to non-curved finite elements.

The existence of multiple approaches, which are quite different from a mathematical point of view, makes it difficult to navigate the literature for those in need of implementing the mapping of a gradient matrix to higher dimensions. This study provides a comprehensive discussion of the three approaches. It is shown that all approaches yield the exact same result. It is illustrated that the third approach can be applied to curved elements by a minor adjustment. Although, rarely discussed, this study highlights that the right Penrose-Moore inverse is an elegant alternative approach to find the gradient matrix in global coordinates.

The mapping of locally defined hydraulic conductivity tensors to the global space is also discussed. Although this mapping is discussed in existing literature for non-curved elements (Kolditz and Glenn, 2002), here a more general mapping is presented that is also applicable to curved fracture elements. This is useful, as such a mapping for curved elements is not discussed in existing literature.

## 2 Preliminary on the geometry of a fracture finite element

Figure 1 illustrates a curved quadrilateral fracture finite element. The orientation of the fracture element can be defined by the normal, strike and dip directions. The local space within the curved quadrilateral is defined by local coordinates $s^k$ with $-1 \le s^k \le 1$. To describe this curved space, some differential geometry of surfaces is needed (Farrashkhalvat and Miles, 2003; Nguyen-SchäFer and Schmidt, 2014; Lebedev et al., 2010; Itskov, 2007). The covariant base vectors are tangent to the local coordinate axes and are given by:

$$\mathbf{a}_k = \frac{\partial x^j}{\partial s^k} \mathbf{e}_j \tag{1}$$

The contravariant base vectors $\mathbf{a}^k$ are perpendicular to planes along which $\mathbf{s}^k$ varies and are given by:

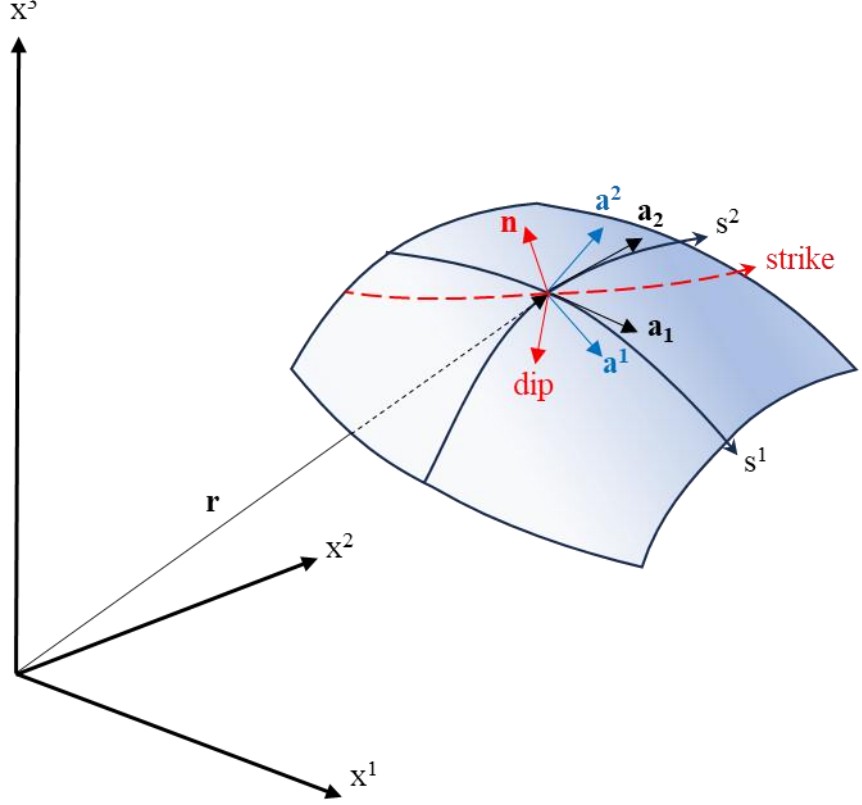

**Figure 1: Geometry of a curved fracture element**

$$\mathbf{a}^k = \frac{\partial s^k}{\partial x^i}\mathbf{e}_i \tag{2}$$

such that:

$$\mathbf{a}^j \cdot \mathbf{a}_i = \delta_j^i \tag{3}$$

where $\delta_j^i$ is the Kronecker delta symbol. The contravariant base vectors and the covariant base vectors are related by:

$$\mathbf{a}_i = G_{ij}\mathbf{a}^j$$
$$\mathbf{a}^i = H^{ij}\mathbf{a}_j \tag{4}$$

where $G_{ij}$ and $H^{ij}$ are the covariant and contravariant metric tensor, respectively. These tensors are given by:

$$G_{ij} = \mathbf{a}_i \cdot \mathbf{a}_j$$
$$H^{ij} = \mathbf{a}^i \cdot \mathbf{a}^j = G_{ij}^{-1} \tag{5}$$

The unit normal vector is simply defined by the cross product of the covariant base vectors:

$$\mathbf{n} = \frac{\mathbf{a}_1 \times \mathbf{a}_2}{|\mathbf{a}_1 \times \mathbf{a}_2|} \tag{6}$$

Making use of Lagrange's identity, the area spanned by the covariant base vectors can be shown to equal the square root of the determinant of $\mathbf{G}$:

$$|\mathbf{a}_1 \times \mathbf{a}_2| = \sqrt{(\mathbf{a}_1 \cdot \mathbf{a}_1)(\mathbf{a}_2 \cdot \mathbf{a}_2) - (\mathbf{a}_1 \cdot \mathbf{a}_2)^2} = \sqrt{\det \mathbf{G}} \tag{7}$$

The local two-dimensional space can be expanded to a local three-dimensional space with the following base vectors all normal to the fracture surface:

$$\mathbf{a}_3 = \mathbf{a}^3 = \mathbf{n} \tag{8}$$

Then equation (3) implies that the contravariant base vectors can also be expressed as:

$$\mathbf{a}^1 = \frac{1}{\sqrt{g}}(\mathbf{a}_2 \times \mathbf{a}_3)$$
$$\mathbf{a}^2 = \frac{1}{\sqrt{g}}(\mathbf{a}_3 \times \mathbf{a}_1) \tag{9}$$
$$\mathbf{a}^3 = \frac{1}{\sqrt{g}}(\mathbf{a}_1 \times \mathbf{a}_2)$$

where $g = \det \mathbf{G}$. It is noted that covariant and contravariant base vectors as well as metric tensors can similarly be defined for triangular finite elements.

## 3 The basic mapping problem

The finite element formulations for groundwater flow result in element matrices that require the element shape functions and their partial derivatives with respect to global Cartesian coordinates. These matrices also involve an integration over the finite element domain $\Omega_e$. For the objective of this study, it suffices to consider the element conductance matrix for saturated groundwater flow:

$$\mathbf{G} = \int_{\Omega_e} \nabla \mathbf{N} \mathbf{K} \nabla \mathbf{N}^T \, d\Omega_e \tag{10}$$

where $\mathit{K}$ is the hydraulic conductivity tensor defined with respect to a global Cartesian coordinate system $x^i$ and $\nabla \mathbf{N}$ the gradient matrix often denoted by $\mathbf{B}$ (Perrochet, 1995):

$$B_{ni} = \frac{\partial N_n}{\partial x^i} \tag{11}$$

where $N_n$ is the $n^{\text{th}}$ nodal shape function. Typically, however, the shape functions are provided with respect to a local coordinate system $s^k$. To find the partial derivatives of the shape functions with respect to global coordinates, the standard approach is to use the Jacobian matrix of the coordinate transformation between local and global space. Following the chain rule:

$$\frac{\partial N_n}{\partial s^k} = \frac{\partial N_n}{\partial x^i} \frac{\partial x^i}{\partial s^k} \tag{12}$$

the Jacobian is defined as follows:

$$J_{ki} = \frac{\partial x^i}{\partial s^k} \tag{13}$$

The components of the Jacobian are computed using the derivates of the shape functions with respect to local coordinates and the nodal coordinates:

$$\frac{\partial x^i}{\partial s^k} = \frac{\partial N_n}{\partial s^k} x_n^i \tag{14}$$

It can be observed that the Jacobian contains the covariant base vectors per row and equation (14) illustrates how to compute these vectors from local shape functions and nodal coordinates. If the Jacobian is invertible, then the derivatives with respect to global coordinates can be computed as follows:

$$\mathbf{B}^{\text{T}} = \mathbf{J}^{-1} \mathbf{B}^{*\text{T}} \tag{15}$$

where $\mathbf{B}^*$ denoted the gradient matrix with respect to local coordinates:

$$B_{nk}^* = \frac{\partial N_n}{\partial s^k} \tag{16}$$

Once $\mathbf{B}^{\text{T}}$ has been computed, the matrix $\mathbf{B}$ can be computed easily by taking the transpose of $\mathbf{B}^{\text{T}}$. Introducing the coordinate matrix $\mathbf{X}$ containing the nodal coordinates per row, it follows from equation (14) that the Jacobian can be computed using:

$$\mathbf{J} = \mathbf{B}^{*T}\mathbf{X} \tag{17}$$

Typically, the element matrices are computed using Gaussian quadrature, although for a limited number of element types, the integration can be carried out analytically (Diersch, 2013). The advantage of numerical integration is that it can be applied to any element type, including curved elements. To perform Gaussian quadrature, the integration limits need to be defined with respect to the local domain $d\Omega^*$. If the Jacobian is invertible, then (Perrochet, 1995):

$$d\Omega = \det(\mathbf{J})d\Omega^* \tag{18}$$

However, if the Jacobian is not a square matrix, then the Jacobian matrix it is not invertible and equation (15) and (18) cannot be used for the finite element computations. This occurs when the local space has a lower dimension than the global space. Thus, for two-dimensional fracture elements embedded within a three-dimensional model space, the problem is that the Jacobian is not a square matrix.

In equation (10), the hydraulic conductivity tensor for fractures is to be defined with respect to the global Cartesian space. In general, it is more convenient to start with tensors which are defined with respect to the strike and dip direction along a fracture. The strike, dip and normal directions provide a locally orthogonal coordinate system. On curvilinear elements, this local coordinate system varies from point to point.

## 4 Gradient mapping using contravariant and covariant bases

Similar to equation (12), it follows from the chain rule that:

$$\frac{\partial N_n}{\partial x^i} = \frac{\partial N_n}{\partial s^k}\frac{\partial s^k}{\partial x^i} \tag{19}$$

This indicates that the gradient matrix with respect to global coordinates can be obtained using the contravariant base vectors. Introducing a matrix $\mathbf{D}$ in which the columns contain the contravariant base vectors:

$$D_{ik} = \frac{\partial s^k}{\partial x^i} \tag{20}$$

it follows that:

$$\nabla \mathbf{N}^T = \mathbf{D}\nabla^*\mathbf{N}^T \tag{21}$$

The components in matrix $\mathbf{D}$ can be rewritten in terms of covariant vectors using equation (5):

$$D_{ij} = \left(\mathbf{a}^j\right)_i = \left(H^{jk}\mathbf{a}_k\right)_i \tag{22}$$

Since the Jacobian $\mathbf{J}$ contains the covariant vectors per row, this can be written as:

$$\mathbf{D} = \mathbf{J}^T \mathbf{H} \tag{23}$$

The contravariant metric tensor $\mathbf{H}$ can also be written in terms of the Jacobian matrices:

$$\mathbf{H} = \mathbf{G}^{-1} = \left(\mathbf{J}\mathbf{J}^T\right)^{-1} \tag{24}$$

where it is noted $\mathbf{J}\mathbf{J}^T$ is an invertible square matrix. Thus, the gradient matrix in global coordinates is given by:

$$\nabla \mathbf{N}^{\mathrm{T}} = \mathbf{J}^T (\mathbf{J}\mathbf{J}^T)^{-1} \nabla^* \mathbf{N}^{\mathrm{T}} \tag{25}$$

Once the Jacobian is available, equation 25 provides a straightforward solution for the gradient matrix in global coordinates. The differential volume follows from equation (7):

$$d\Omega = \sqrt{\det(\mathbf{G})}\, d\Omega^* = \sqrt{\det(\mathbf{J}\mathbf{J}^T)}\, d\Omega^* \tag{26}$$

It is interesting to observe that equation (9) permits to write the matrix $\mathbf{D}$ as:

$$\mathbf{D} = \frac{1}{\sqrt{g}} \begin{bmatrix} (\mathbf{a}_2 \times \mathbf{a}_3)_1 & (\mathbf{a}_3 \times \mathbf{a}_1)_1 \\ (\mathbf{a}_2 \times \mathbf{a}_3)_2 & (\mathbf{a}_3 \times \mathbf{a}_1)_2 \\ (\mathbf{a}_2 \times \mathbf{a}_3)_3 & (\mathbf{a}_3 \times \mathbf{a}_1)_3 \end{bmatrix} \tag{27}$$

Using the vector triple product, it can be shown that:

$$\mathbf{a}_2 \times \mathbf{a}_3 = \mathbf{a}_2 \times \frac{\mathbf{a}_1 \times \mathbf{a}_2}{|\mathbf{a}_1 \times \mathbf{a}_2|} = \frac{1}{\sqrt{g}}\left((\mathbf{a}_2 \cdot \mathbf{a}_2)\mathbf{a}_1 - (\mathbf{a}_2 \cdot \mathbf{a}_1)\mathbf{a}_2\right)$$

$$\mathbf{a}_3 \times \mathbf{a}_1 = \frac{\mathbf{a}_1 \times \mathbf{a}_2}{|\mathbf{a}_1 \times \mathbf{a}_2|} \times \mathbf{a}_1 = \frac{1}{\sqrt{g}}\left((\mathbf{a}_1 \cdot \mathbf{a}_1)\mathbf{a}_2 - (\mathbf{a}_1 \cdot \mathbf{a}_2)\mathbf{a}_1\right) \tag{28}$$

Eventually, after expanding the cross products in equation (27) using the vector triple products in equation (28), it can be shown that this eventually yield the same result $\mathbf{D} = \mathbf{J}^T (\mathbf{J}\mathbf{J}^T)^{-1}$.

## 5 Gradient mapping using the right Penrose-Moore inverse

Equation (15) can be written as:

$$\mathbf{J}\nabla \mathbf{N}^{\mathrm{T}} = \nabla^* \mathbf{N}^{\mathrm{T}} \tag{29}$$

Since the Jacobian is rectangular, equation (29) represents an underdetermined system with infinite many solutions. However, the particular solution that represents the desired mapping needs to be a solution that lies in the row space of $\mathbf{J}$. Namely, the row space of $\mathbf{J}$ contains the covariant base vectors spanning the local fracture space. To reflect this condition, equation (29) is written as:

$$\mathbf{J}\mathbf{J}^T\mathbf{M} = \nabla^*\mathbf{N}^T \tag{30}$$

where the matrix $\nabla\mathbf{N}^T = \mathbf{J}^T\mathbf{M}$ now lies within the row space of $\mathbf{J}$. Equation (30) has a unique solution:

$$\mathbf{M} = \left(\mathbf{J}\mathbf{J}^T\right)^{-1}\nabla^*\mathbf{N}^T \tag{31}$$

Thus, the same result as in equation (25) is obtained:

$$\nabla\mathbf{N}^T = \mathbf{J}^T\left(\mathbf{J}\mathbf{J}^T\right)^{-1}\nabla^*\mathbf{N}^T \tag{32}$$

This can also be written as:

$$\nabla\mathbf{N}^T = \mathbf{J}^\dagger\nabla^*\mathbf{N}^T \tag{33}$$

where $\mathbf{J}^\dagger$ is the so-called right Penrose-Moore inverse given by:

$$\mathbf{J}^\dagger = \mathbf{J}^T\left(\mathbf{J}\mathbf{J}^T\right)^{-1} \tag{34}$$

A more in-depth background on the Penrose-Moore inverse is provided in the Appendix.

**6 Gradient mapping using directional cosines**

For each point on a possibly curved two-dimensional discrete element, it is possible to construct a two-dimensional orthonormal coordinate system tangent to the fracture defined by unit vectors $\hat{\mathbf{e}}_1$ and $\hat{\mathbf{e}}_2$. There are several possibilities, but here the procedure starts with taking the vector $\hat{\mathbf{e}}_1$ parallel to the first covariant basis $\mathbf{a}_1$:

$$\hat{\mathbf{e}}_1 = \frac{\mathbf{a}_1}{|\mathbf{a}_1|} \tag{35}$$

The vector $\hat{\mathbf{e}}_2$ can be easily obtained making use of the normal $\mathbf{n}$.

$$\hat{\mathbf{e}}_2 = \hat{\mathbf{e}}_1 \times \mathbf{n} \tag{36}$$

This two-dimensional orthonormal coordinate system can be expanded into three dimensions by adding a third unit vector:

$$\hat{\mathbf{e}}_3 = \mathbf{n} \tag{37}$$

The differential volume simply follows from the covariant base vectors:

$$d\Omega = \left|\mathbf{a}_1 \times \mathbf{a}_2\right| d\Omega^* \tag{38}$$

The transformation from to the global coordinate system to the new coordinate system $\hat{x}^i$ is given by a 2 by 3 matrix of directional cosines:

$$T_{ij}^{2x3} = \frac{\partial \hat{x}^i}{\partial x^j} = \hat{\mathbf{e}}_i \cdot \mathbf{e}_j = \cos(\hat{x}^i, x^j) \tag{39}$$

The matrix of directional cosines can also be expressed as:

$$\mathbf{T} = \begin{bmatrix} \hat{\mathbf{e}}^1 \\ \hat{\mathbf{e}}^2 \end{bmatrix} \tag{40}$$

where it is noted that the inverse of $\mathbf{T}$ is $\mathbf{T}^T$.

The gradient matrix with respect to the new two-dimensional orthonormal coordinate system $\nabla^\wedge \mathbf{N}^T$ is given by:

$$\nabla^\wedge \mathbf{N}^T = \hat{\mathbf{J}}^{-1} \nabla^* \mathbf{N}^T \tag{41}$$

where the Jacobian matrix $\hat{\mathbf{J}}$ is an invertible 2 by 2 matrix:

$$\hat{J}_{ki} = \frac{\partial \hat{x}^i}{\partial s^k} = \frac{\partial N_n}{\partial s^k} \hat{x}_n^i \tag{42}$$

Using the global coordinate matrix, this Jacobian is computed with:

$$\hat{\mathbf{J}} = \mathbf{B}^{*T} \mathbf{X} \mathbf{T}^T \tag{43}$$

The gradient matrix with respect to global coordinates follows from using the chain rule:

$$\frac{\partial N_n}{\partial x^i} = \frac{\partial N_n}{\partial \hat{x}^k} \frac{\partial \hat{x}^k}{\partial x^i} \tag{44}$$

which can be expressed using the transformation matrix:

$$\nabla \mathbf{N}^T = \mathbf{T}^T \hat{\mathbf{J}}^{-1} \nabla^* \mathbf{N}^T \tag{45}$$

This expression looks quite different compared from the expressions obtained using the first and second approach. However, it can be illustrated that the result is identical. From the chain rule:

$$\frac{\partial s^k}{d\hat{x}^i} = \frac{\partial x^j}{d\hat{x}^i}\frac{\partial s^k}{dx^j} \tag{46}$$

it follows that:

$$\hat{\mathbf{J}}^{-1} = \mathbf{T}\mathbf{D} \tag{47}$$

Therefore, the equation 49 is identical to equation 21:

$$\nabla\mathbf{N}^T = \mathbf{T}^T\hat{\mathbf{J}}^{-1}\nabla^*\mathbf{N}^T = \mathbf{T}^T\mathbf{T}\mathbf{D}\nabla^*\mathbf{N}^T = \mathbf{D}\nabla^*\mathbf{N}^T \tag{48}$$

Since the covariant bases are used to construct a two-dimensional orthonormal coordinate system, the approach as discussed here is applicable to curved fracture elements. In existing literature (Diersch, 2013; Kolditz and Glenn, 2002; Watanabe, 2011), the two-dimensional orthonormal space is often constructed using the edges of non-curved fracture elements. That is, the unit normal is constructed from two element edges, the first unit vector is taken parallel to the first edge and finally a cross product of the unit normal and the first unit vector is used to compute the second unit vector. Such an approach assumes that the two-dimensional orthonormal space is constant across the fracture element, which is only valid for non-curved fracture elements.

## 7 Coordinate transformations for hydraulic conductivity tensor

Here, it is assumed that a hydraulic tensor is initially provided with respect to the local strike and dip directions for each fracture element. On curvilinear elements, the strike and dip directions vary from point to point. Given the normal $\mathbf{n}$, which also varies from point to point within a curved fracture element and a vertical unit vector $\mathbf{v}$, the unit vector in the strike direction is given by:

$$\widehat{\mathbf{e}}_1 = \mathbf{n} \times \mathbf{v} \tag{49}$$

The unit vector in the dip direction follows directly from the following cross product:

$$\widehat{\mathbf{e}}_2 = \mathbf{n} \times \widehat{\mathbf{e}}_1 \tag{50}$$

Finally, the unit vector normal to the fracture is given by:

$$\widehat{\mathbf{e}}_3 = \mathbf{n} \tag{51}$$

The transformation from the orthonormal local coordinate system aligned with the strike and dip direction to the global coordinate system is defined by the following 3 by 2 matrix:

$$\mathbf{Q} = \begin{bmatrix} \hat{\mathbf{e}}^1 & \hat{\mathbf{e}}^2 \end{bmatrix} \tag{52}$$

Denoting the two-dimensional hydraulic conductivity tensor in local coordinates by $\hat{\mathbf{K}}$, the hydraulic conductivity tensor in global coordinates is given by:

$$\mathbf{K} = \mathbf{Q}\hat{\mathbf{K}}\mathbf{Q}^T \tag{53}$$

For curved elements the normal is to be computed from the covariant vectors using equation (6). For non-curved elements, the normal is constant across the element and can be computed by taking the cross-product between two element edges.

## 8 Example

To illustrate how the different gradient mappings are applied in practice, a curved quadratic triangular element is considered. Three Gauss points, each at the midpoint on an edge, are used for numerical integration. The nodal shape functions are defined by (OñAte, 2010):

$$\begin{aligned}
N_1 &= (1-r-s)(1-2r-2s) \\
N_2 &= r(2r-1) \\
N_3 &= s(2s-1) \\
N_4 &= 4r(1-r-s) \\
N_5 &= 4rs \\
N_6 &= 4s(1-r-s)
\end{aligned} \tag{54}$$

with $r$ and $s$ the local coordinates ( $0 \le r \le 1$ and $0 \le s \le 1$ ). The gradient matrix with respect to local coordinates, which varies within the triangle, is given by:

$$\mathbf{B}^* = \begin{bmatrix}
-3+4r+4s & -3+4r+4s \\
4r-1 & 0 \\
0 & 4s-1 \\
4-8r-4s & -4r \\
4s & 4r \\
-4s & 4-4r-8s
\end{bmatrix} \tag{55}$$

In this example, the coordinates of the element nodes are given by the coordinate matrix:

$$\mathbf{X} = \begin{bmatrix} 0 & 0 & 0 \\ 1 & 0 & 0 \\ 0 & 0 & 1 \\ \dfrac{1}{2} & 0 & 0 \\ \dfrac{1}{2} & -\dfrac{1}{8} & \dfrac{1}{2} \\ 0 & -\dfrac{1}{8} & \dfrac{1}{2} \end{bmatrix} \qquad (56)$$

Figure 2 illustrates the triangle.

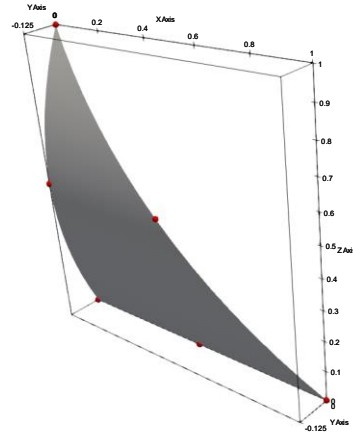

**Figure 2: The curved quadratic triangle as used in the example. Nodes represented as red circles.**

The conductivity tensor in this example is defined as:

$$\hat{\mathbf{K}} = \begin{bmatrix} 1 & 0 \\ 0 & 1 \end{bmatrix} \qquad (57)$$

The Jacobian for each Gauss point is computed using $\mathbf{J} = \mathbf{B}^{*T}\mathbf{X}$. For the first Gauss point given by $(r,s)=(0,1/2)$, the Jacobian

is:

$$\mathbf{J} = \begin{bmatrix} 1 & 0 & 0 \\ 0 & -\dfrac{1}{2} & 1 \end{bmatrix} \tag{58}$$

This Jacobian provides the covariant base vectors per row. Once the Jacobian is computed, the gradient matrix in global coordinates is easily computed using $\nabla \mathbf{N}^{\mathrm{T}} = \mathbf{J}^T (\mathbf{JJ}^T)^{-1} \nabla^* \mathbf{N}^{\mathrm{T}}$. This is the expression that results from either using contravariant and covariant bases or a right Penrose-Moore inverse. For the first Gauss point the gradient matrix is given by:

$$\nabla \mathbf{N}^{\mathrm{T}} = \begin{bmatrix} -1 & 1 & 0 & 0 & 0 & 0 \\ \dfrac{2}{5} & 0 & \dfrac{2}{5} & \dfrac{4}{5} & -\dfrac{4}{5} & -\dfrac{4}{5} \\ -\dfrac{4}{5} & 0 & -\dfrac{4}{5} & -\dfrac{8}{5} & \dfrac{8}{5} & \dfrac{8}{5} \end{bmatrix} \tag{59}$$

and:

$$\partial\Omega = \sqrt{\det(\mathbf{JJ}^T)} = \frac{\sqrt{5}}{2} \tag{60}$$

The transformation matrix for the hydraulic conductivity tensor is obtained from the vectors $\hat{\mathbf{e}}_1$ and $\hat{\mathbf{e}}_1$. For the first Gauss point:

$$\mathbf{Q} = \begin{bmatrix} 0 & -1 \\ -\dfrac{\sqrt{5}}{5} & 0 \\ \dfrac{2\sqrt{5}}{5} & 0 \end{bmatrix} \tag{61}$$

Applying equation 53 gives:

$$\mathbf{K} = \begin{bmatrix} 1 & 0 & 0 \\ 0 & \dfrac{1}{5} & \dfrac{-2}{5} \\ 0 & \dfrac{-2}{5} & \dfrac{4}{5} \end{bmatrix} \tag{62}$$

The contribution to **G** from each gauss point $i$ is computed from:

$$w_i \left( \nabla \mathbf{N} \mathbf{K} \nabla \mathbf{N}^T \partial \Omega \right)_i \tag{63}$$

where $w_i$ is the gauss weight for each Gauss point $i$. For the triangular element considered here, this weight is 1/6 for all gauss points (Diersch, 2013; OñAte, 2010). For the first Gauss point, this contribution equals:

$$\begin{bmatrix}
\dfrac{3}{20} & \dfrac{-1}{12} & \dfrac{1}{15} & \dfrac{2}{15} & \dfrac{-2}{15} & \dfrac{-2}{15} \\[2mm]
\dfrac{-1}{12} & \dfrac{1}{12} & 0 & 0 & 0 & 0 \\[2mm]
\dfrac{1}{15} & 0 & \dfrac{1}{15} & \dfrac{2}{15} & \dfrac{-2}{15} & \dfrac{-2}{15} \\[2mm]
\dfrac{2}{15} & 0 & \dfrac{2}{15} & \dfrac{4}{15} & \dfrac{-4}{15} & \dfrac{-4}{15} \\[2mm]
\dfrac{-2}{15} & 0 & \dfrac{-2}{15} & \dfrac{-4}{15} & \dfrac{4}{15} & \dfrac{4}{15} \\[2mm]
\dfrac{-2}{15} & 0 & \dfrac{-2}{15} & \dfrac{-4}{15} & \dfrac{4}{15} & \dfrac{4}{15}
\end{bmatrix} \sqrt{5} \tag{64}$$

For brevity, the calculations for the other two Gauss points are not presented, but are computed similarly.

Using the gradient mapping based on directional cosines requires more work, as illustrated below. After obtaining the Jacobian, the covariant base vectors are used to compute $\hat{\mathbf{e}}_1$ and $\hat{\mathbf{e}}_2$. For the first Gauss point, this results in the following 2 by 3 transformation matrix:

$$\mathbf{T} = \begin{bmatrix} 1 & 0 & 0 \\[2mm] 0 & \dfrac{\sqrt{5}}{5} & \dfrac{-2\sqrt{5}}{5} \end{bmatrix} \tag{65}$$

Using $\hat{\mathbf{J}} = \mathbf{B}^{*T} \mathbf{X} \mathbf{T}^T$ (equation 43) and $\nabla \mathbf{N}^T = \mathbf{T}^T \hat{\mathbf{J}}^{-1} \nabla^* \mathbf{N}^T$ (equation 45) then yields the same result as before (equation 60).

## 9 Application

The application considers steady-state flow towards a pipe penetrating a curved fracture using quadratic triangles. Although relatively simple this model application could be used for example to simulate the water inrush during tunnel construction when a fracture is penetrated, provided the geometry and hydraulic properties of the fracture are known. The triangular mesh is generated with Triangle (Shewchuk, 1996) on a flat surface with a length of a 100 m and a height of 50 m. Subsequently, the mesh is wrapped on a curved surface. The pipe is represented by a hole approximating a circular pipe that intersects the fracture horizontally. A constant pressure head of 0 m is prescribed at the nodes around this hole. At the top and bottom of the fracture, constant hydraulic heads of 10 m are prescribed. The hydraulic conductivity tensor is given by:

$$\widehat{\mathbf{K}} = \begin{bmatrix} 0.75 & 0.25 \\ 0.25 & 0.75 \end{bmatrix} 10^{-5}\,\text{m/s} \tag{66}$$

Thus, the principal axes of this tensor make an angle of 45 degrees with respect to local strike and dip directions.

The simulation is carried out with a finite element groundwater flow model developed by the author. Figure 3 illustrates the solution in terms of hydraulic heads inside the fracture. It is observed that the hydraulic head contours in the vicinity of the pipe follow an ellipsoid as a result of the anisotropic hydraulic conductivity tensor. The pipe penetrating the fracture is represented by a hole instead of a single boundary node for two reasons. Firstly, by using a hole, the radius of the pipe is accounted for (approximately). Secondly, since the hole is represented by relatively short edges, relatively small triangles are generated around the pipe, such that the model can better resolve the steep hydraulic gradients around this part of the model domain.

It is noticed that because the fracture does not lie in a flat plane, the fracture is embedded in three-dimensional space. Therefore, gradient matrices must be mapped from two-dimensional local to three-dimensional global space. In the finite element code applied here, this mapping is preferably based the expression $\nabla \mathbf{N}^{\text{T}} = \mathbf{J}^{T}(\mathbf{J}\mathbf{J}^{T})^{-1}\nabla^{*}\mathbf{N}^{\text{T}}$ as the routines for this mapping are most concise. The alternative gradient mapping based on directional cosines provides exactly the same result. The hydraulic conductivity tensor must also be mapped to global space.

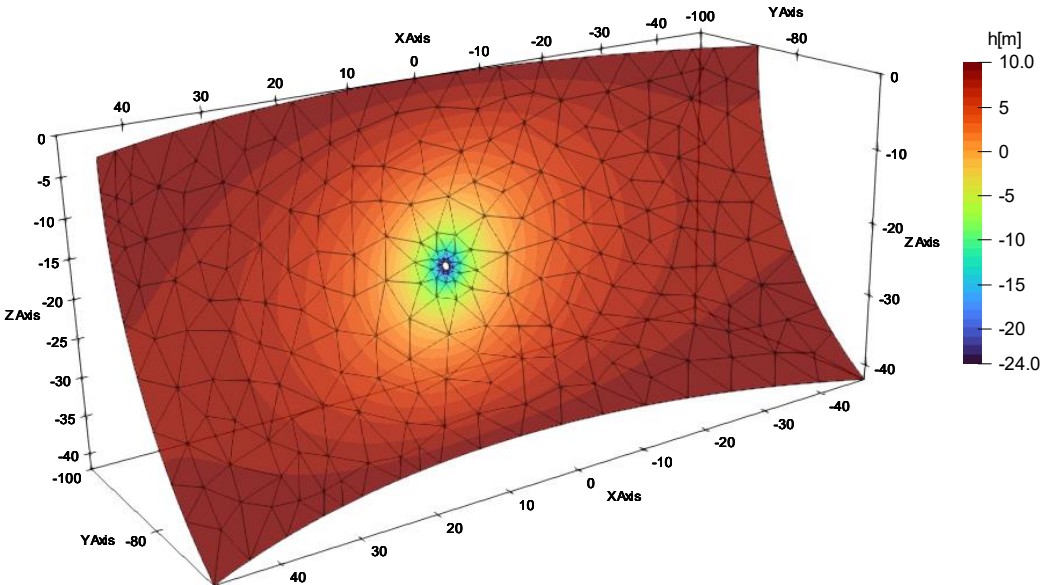

**Figure 3: Hydraulic heads simulated in a curved fracture with an anisotropic hydraulic conductivity tensor.**

## 10 Discussion and conclusion

A key component in the finite element method is the mapping of the gradient matrix from local to global space. If the global space has the same dimension as the local space, then the mapping is defined by the inverse of the Jacobian matrix. However,

in the case of two-dimensional fractures embedded in a global three-dimensional space, the Jacobian matrix is non-square and thus not invertible (Juanes et al., 2002; Perrochet, 1995). A couple of different techniques enable a mapping from two-dimensional local to three-dimensional global space.

It is shown in this work that applying the right Penrose-Moore inverse is an efficient, elegant and relatively simple alternative to find an expression to map the gradient matrix. This alternative avoids the use of tensor calculus or the use of

cumbersome rotation matrices. Instead, it uses the concept of subspaces associated with matrices. It is also shown that the mapping approach based on an intermediate mapping to a two-dimensional orthonormal space and a subsequent mapping to the global space can be applied to curved elements. The approach based on the right Penrose-Moore inverse, the approach based on covariant and contravariant vectors and the approach based on an intermediate mapping to a two-dimensional orthonormal space and a subsequent mapping to the global space all yield the same mapping result. If the Jacobian $\mathbf{J}$ is readily

available, the expression $\nabla \mathbf{N}^{\mathrm{T}} = \mathbf{J}^{T}(\mathbf{J}\mathbf{J}^{T})^{-1}\nabla^{*}\mathbf{N}^{\mathrm{T}}$ as derived from the right Penrose-Moore inverse or from the approach based on covariant and contravariant vectors is particularly straightforward to implement in a computer code. While the approach

based on an intermediate mapping to a two-dimensional orthonormal space and a subsequent mapping to the global space is easier to understand from a mathematical point of view, it involves extra steps that involve cumbersome rotation matrices. The Jacobian **J** is readily available if the finite element code uses numerical integration. It is noted that analytical integration avoids

the need to define the local space and as such the Jacobian **J** is not defined. Thus, if analytical integration is used, then there is no alternative for implementing the intermediate mapping to a two-dimensional orthonormal space and the subsequent rotation to the global space. In general, however, it can be argued that numerical integration is to be preferred, since it is far easier to implement (even without considering the mapping problem for fracture elements). Moreover, numerical integration is more general as it can be applied to all finite element types including curved elements.

Finally, this work provides a general approach, applicable to curved elements, to map hydraulic tensors as defined in a local orthonormal coordinate system aligned with the strike, dip and normal directions to the global coordinate system.

**Appendix**

The Penrose-Moore inverse is widely used to solve over-determined and under-determined linear systems. By definition, the Penrose-Moore inverse satisfies the following conditions (Penrose, 1955):

$$
\begin{aligned}
\mathbf{A}\mathbf{A}^{\dagger}\mathbf{A} &= \mathbf{A} & (I) \\
\mathbf{A}^{\dagger}\mathbf{A}\mathbf{A}^{\dagger} &= \mathbf{A} & (II) \\
\mathbf{A}\mathbf{A}^{\dagger} &= \left(\mathbf{A}\mathbf{A}^{\dagger}\right)^{T} & (III) \\
\mathbf{A}^{\dagger}\mathbf{A} &= \left(\mathbf{A}^{\dagger}\mathbf{A}\right)^{T} & (IV)
\end{aligned}
\tag{67}
$$

Condition (I) implies that $\mathbf{A}^{\dagger}\mathbf{A}$ is idempotent ( $\mathbf{A}^{\dagger}\mathbf{A}\mathbf{A}^{\dagger}\mathbf{A} = \mathbf{A}^{\dagger}\mathbf{A}$ ) and condition (IV) implies that $\mathbf{A}^{\dagger}\mathbf{A}$ is hermetian. Therefore $\mathbf{A}^{\dagger}\mathbf{A}$ is an orthogonal projection matrix. Using the right Penrose-Moore inverse $\mathbf{A}^{\dagger} = \mathbf{A}^{T}\left(\mathbf{A}\mathbf{A}^{T}\right)^{-1}$ as used for an under-determined system, $\mathbf{A}^{\dagger}\mathbf{A}$ is expressed as:

$$
\mathbf{A}^{\dagger}\mathbf{A} = \mathbf{A}^{T}\left(\mathbf{A}\mathbf{A}^{T}\right)^{-1}\mathbf{A}
\tag{68}
$$

This last expression illustrates that $\mathbf{A}^{\dagger}\mathbf{A}$ is the orthogonal projection matrix $\mathbf{P}_{\mathcal{R}\left(A^{T}\right)}$ onto the column space or range of $\mathbf{A}^{T}$ (Strang, 2022) which equals the row space or range of $\mathbf{A}$ and as such $\mathbf{I} - \mathbf{A}\mathbf{A}^{\dagger}$ is the orthogonal projection matrix $\mathbf{P}_{\mathcal{N}(A)}$ onto the nullspace of $\mathbf{A}$:

$$
\begin{aligned}
\mathbf{P}_{\mathcal{R}\left(A^{T}\right)} &= \mathbf{A}^{\dagger}\mathbf{A} \\
\mathbf{P}_{\mathcal{N}(A)} &= \mathbf{I} - \mathbf{A}^{\dagger}\mathbf{A}
\end{aligned}
\tag{69}
$$

Using these orthogonal projection matrices, a solution to an under-determined system $\mathbf{A}\mathbf{x}=\mathbf{b}$ can thus be expressed as:

$$\mathbf{x} = \left(\mathbf{A}^{\dagger}\mathbf{A}\right)\mathbf{x} + \left(\mathbf{I} - \mathbf{A}^{\dagger}\mathbf{A}\right)\mathbf{x} = \mathbf{A}^{\dagger}\mathbf{b} + \left(\mathbf{I} - \mathbf{A}^{\dagger}\mathbf{A}\right)\mathbf{x} \tag{70}$$

This illustrates that the right Penrose-Moore inverse provides the solution $\mathbf{x} = \mathbf{A}^{\dagger}\mathbf{b}$ that lies within the row space of $\mathbf{A}$.

**Code availability**

The code used in the application can be obtained from the author

**Competing interests**

The contact author declares no competing interests

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
