# Peer review of "Technical note: Finite element formulations to map discrete fracture elements in three-dimensional groundwater models"

_Hydrology and Earth System Sciences, 2024_

## Author Response (AR2)

**RC1**

This manuscript presents a methodology for including two-dimensional elements, such as fractures, in a three-dimensional porous media matrix. The authors claim that the proposed methodology is easier to implement than the commonly adopted method composed of a two-dimensional mapping on an orthonormal space, followed by a three-dimensional rotation. In addition, the proposed methodology works for curved two-dimensional surfaces with the integration performed using the Gaussian quadrature method, a case that cannot be addressed with the existing methodology. The proposed new approach cannot be applied when the integration is performed analytically (in cases where this is possible).

The technical note is "dry" in the sense that it discusses only the new projection method showing how it works only for one of the integrals (Eq. 10) that should be computed in the FEM solution of the flow equation. The discussion on the hydraulic conductivity projection of the 2-D elements (the fractures) presented in Section 6 is appreciable, but the manuscript lacks an illustrative example, which would be certainly appreciated by the HESS readership and myself too.

The proposed methodology is sound. However, as mentioned above, the presentation is somewhat "dry," and including an application example would enhance the reader's understanding of the benefits offered by the proposed approach. The example does not need to be geometrically complex; for instance, a cubic homogeneous formation with a single curvilinear fracture, an imposed head gradient between two opposing faces, and the remaining faces set as impervious would be sufficient to demonstrate the methodology's capabilities. In addition, I suggest that the authors compare the solution obtained using their proposed approach with that obtained through the existing and widely adopted projection and rotation methodology.

**RC2**

The authors propose several approaches to map local spaces of fractures embedded in higher-dimensional domain. My main concern is related to the fact that the proposed approaches should be numerically validate, showing their performances for complex fracture networks. This makes the reader difficult to evaluate which algorithm should be preferred. I have also the following minor comments:

- eq (3) use a different symbol since it resembles a partial derivative

- the part on the Penrose-Moore inverse can be moved to an appendix

**Changes to manuscript:**

- I added 2 sections (before the discussion)
  1) An example to illustrate the actual implementation using a single curved element. This example also shows that the third mapping approach requires a bit more work. For brevity,

the example is not worked out completely in all its details, but provides enough information for those interested in implantation details.

2)   An application that requires the kind of mappings discussed in this manuscript.

Both sections, but particularly the last section are added because both reviewers requested a simulation example. Here I provide an example on a curved surface that uses quadratic triangular elements. This is the same kind of element that I discuss in the example. Since it is a steady-state problem, it only involves the conductance matrix.

I tried to provide an application that could be useful (potentially) for a practical problem. The application here could be useful to simulate water inrush during tunnel construction when a fracture is crossed. I also tried to provide an example that corresponds to the premises in the theoretical part. Hence, I used curved elements in the application.

- As proposed by Reviewer 2, I moved some details about the Penrose-Moore matrix to an Appendix.
- I corrected the symbol used for the Kronecker delta (equation 3)
- I added a few equations in anticipation of the example (17 and 43). In terms of how to implement the mappings, I think that these two additions are quite useful (in a practical sense).
- In section 6 (Gradient mapping using directional cosines), I made a few corrections. Those affect the last equations.
- An additional reference has been added which is used in the example.

I would like to thank the reviewers for their comments. I agree that the application example is a useful addition to the manuscript. In addition, I decided to provide an example showing how the mappings are implemented by just looking at a single element.

Rob de Rooij